# Life course exposures continually shape antibody profiles and risk of seroconversion to influenza

Bingyi Yang[1,2]*, Justin Lessler[3], Huachen Zhu[4,5], Chao Qiang Jiang[6], Jonathan M. Read[7], James A. Hay[8¤], Kin On Kwok[9,10,11], Ruiyin Shen[6], Yi Guan[4,5], Steven Riley[8]*, Derek A. T. Cummings[1,2]*

1 Department of Biology, University of Florida, Gainesville, Florida, United States of America, 2 Emerging Pathogens Institute, University of Florida, Gainesville, Florida, United States of America, 3 Department of Epidemiology, Johns Hopkins Bloomberg School of Public Health, Baltimore, Maryland, United States of America, 4 State Key Laboratory of Emerging Infectious Diseases and Centre of Influenza Research, School of Public Health, The University of Hong Kong, Hong Kong SAR, China, 5 Joint Institute of Virology (Shantou University–The University of Hong Kong), Shantou University, Shantou, Guangdong, China, 6 Guangzhou No.12 Hospital, Guangzhou, Guangdong, China, 7 Centre for Health Informatics Computing and Statistics, Lancaster Medical School, Lancaster University, Lancaster, United Kingdom, 8 MRC Centre for Global Infectious Disease Analysis, Department of Infectious Disease Epidemiology, School of Public Health, Imperial College London, London, United Kingdom, 9 The Jockey Club School of Public Health and Primary Care, The Chinese University of Hong Kong, Hong Kong Special Administrative Region, China, 10 Stanley Ho Centre for Emerging Infectious Diseases, The Chinese University of Hong Kong, Shatin, Hong Kong Special Administrative Region, China, 11 Shenzhen Research Institute of The Chinese University of Hong Kong, Shenzhen, Guangdong, China

¤ Current address: Center for Communicable Disease Dynamics, Department of Epidemiology, Harvard T. H. Chan School of Public Health, Boston, Massachusetts, United States of America
* yangb@ufl.edu (BY); s.riley@imperial.ac.uk (SR); datc@ufl.edu (DATC)

**Data Availability Statement:** All relevant data and code used to reproduce the study findings are available at (https://github.com/UF-IDD/Fluscape_Paired_Serology).

## Abstract

Complex exposure histories and immune mediated interactions between influenza strains contribute to the life course of human immunity to influenza. Antibody profiles can be generated by characterizing immune responses to multiple antigenically variant strains, but how these profiles vary across individuals and determine future responses is unclear. We used hemagglutination inhibition titers from 21 H3N2 strains to construct 777 paired antibody profiles from people aged 2 to 86, and developed novel metrics to capture features of these profiles. Total antibody titer per potential influenza exposure increases in early life, then decreases in middle age. Increased titers to one or more strains were seen in 97.8% of participants during a roughly four-year interval, suggesting widespread influenza exposure. While titer changes were seen to all strains, recently circulating strains exhibited the greatest titer rise. Higher pre-existing, homologous titers at baseline reduced the risk of seroconversion to recent strains. After adjusting for homologous titer, we also found an increased frequency of seroconversion against recent strains among those with higher immunity to older previously exposed strains. Including immunity to previously exposures also improved the deviance explained by the models. Our results suggest that a comprehensive quantitative description of immunity encompassing past exposures could lead to improved correlates of risk of influenza infection.

**Funding:** This study was supported by grants from the NIH R56AG048075 (D.A.T.C., J.L.), NIH R01AI114703 (D.A.T.C., B.Y.), the Wellcome Trust 200861/Z/16/Z (S.R.) and 200187/Z/15/Z (S.R.). D. A.T.C., J.M.R. and S.R. acknowledge support from the National Institutes of Health Fogarty Institute (R01TW0008246). J.M.R. acknowledges support from the Medical Research Council (MR/S004793/ 1) and the Engineering and Physical Sciences Research Council (EP/N014499/1). S.R. acknowledges National Institute for Health Research (UK, for Health Protection Research Unit funding). The funders had no role in study design, data collection and analysis, decision to publish, or preparation of the manuscript.

**Competing interests:** The authors have declared that no competing interests exist.

## Author summary

Antibody profiles characterize immunity arising from multiple influenza infections during a lifetime and could provide more information than measuring homologous titers. As antibody profiles consist of complex cross-reactions and are difficult to quantify, how past exposures vary across people and time and determine the risk of future infections remains unclear. Here, we develop several metrics to define the key characteristics of antibody profiles, including the overall levels, the breadth and temporal center of mass. With these metrics, we show that immunity accumulates during the first twenty years of life and then declines until 40–50 years old. This pattern is likely driven by the widespread influenza exposure as we find during the four-year periods. Further, we show that individuals with higher antibody to antigenically distant strains had a higher frequency of seroconversion to recent strains, with an unclear underlying mechanism. Our work provides quantitative tools to analyze complex antibody profiles and improve the understanding of heterogeneity in antibody response and vaccine efficacy across age groups.

## Introduction

Seasonal influenza remains a ubiquitous threat to human health. It is estimated to kill between 291,000 and 645,000 people each year worldwide [1]. Through the process of antigenic drift, antigenically novel strains replace previously circulating viruses every few influenza seasons [2]. As a result, people can be infected multiple times over a lifetime [3]. Each of these infections leaves a mark on a person's immune system, and the accumulation of antibody responses over a life course leads to complex individual antibody profiles reflecting both recent and past exposures [3–6]. A growing body of evidence suggests that the order and timing of influenza exposures shape the immune response in ways that may affect morbidity and mortality [3,7], particularly when encountering novel (i.e. pandemic or potentially pandemic) strains [8], yet a comprehensive quantitative description of how past exposure to multiple strains shapes infection risk remains elusive.

Historically, most studies have focused on measuring antibodies to a single strain or small set of strains of a subtype, either to measure the incidence of influenza through changes in titers [9–11] or as a proxy for immunity (e.g., as an endpoint in vaccine efficacy studies) [12–15]. Exceptions do exist, and studies of multi-strain dynamics after immunological challenge date back to at least 1941 [16,17]. However, historically, few analytical approaches were available with which to interpret these complex data. In recent years there has been increasing interest in measuring and interpreting the breadth and strength of antigenic profiles across a diverse set of influenza strains [4,7]. It is believed to provide a more nuanced picture of the humoral immune response to influenza than measuring titers to single strains (i.e. homologous titers), while the quantitative role of past exposures in determining risk of future infections and subsequent immune responses is still unclear.

Here, we describe paired antibody profiles measured at two time points (baseline from 2009 to 2011 and follow-up from 2014 to 2015), roughly four years apart, in a large sample of individuals from an ongoing cohort study in Guangzhou, Guangdong Province, China [18]. We measured immune responses to multiple chronologically ordered H3N2 influenza strains (referred to as antibody profiles) that represent the history of H3N2 circulation in humans since its emergence in 1968. We aim to determine how those profiles vary across individuals and between study visits, and to test if there exist features of these antibody profiles that are

more predictive of the odds of seroconversion to recently circulating strains than homologous titers only.

Antibody responses to 21 H3N2 influenza strains were measured for each individual from sera using hemagglutination inhibition assays (HAI). The strains selected covered the period between 1968–2014, and included strains isolated at 2–3 year intervals. Strains included in the vaccine formulation and tested by Fonville et al. [7] were prioritized, but when this was not possible another antigenically representative strain from that year was selected. We included two strains that were isolated in 2009 (i.e. A/Perth/2009 and A/Victoria/2009) to address the potential difference in the circulating strain and vaccine strain in that year. We described individual antibody profiles from these serological data, and introduced multiple novel metrics to summarize each individual's profile and facilitate comparisons of profiles across age and time. We also determined if summary metrics of antibody profiles improved predictions of the odds of seroconversion. We define seroconversion against strain as four-fold or greater rise in titers to that strain between baseline and follow-up, which is a commonly used indicator of influenza exposure [14].

## Results

### Serological data and individual antibody profiles

Both baseline (December 4, 2009 to January 22, 2011, with 44% of serum collected after the 2010 summer season started) and follow-up (June 17, 2014 to June 2, 2015, with 94% of serum collected after the 2014 summer season started) sera were available from 777 participants aged from 2 to 86 years old (age at baseline is used throughout unless otherwise noted) [18–20]. Participants giving serum were largely representative of the censused and overall study populations, with under representation of young children and over representation of participants aged 40–59 years (S1 Table) [18]. There were 10 participants (1.3%) who reported being vaccinated against influenza at baseline, and 5 participants (0.6%) who reported influenza vaccination between the two visits.

In total, 32,606 HAI titer readings were available, with only 28 missing out of $21 \times 777 \times 2$ possible strain-individual-visit combinations. Figs 1 and S1 shows example individual antibody profiles for an age representative subset of individuals. We defined the four strains (i.e. A/Perth/2009, A/Victoria/2009, A/Texas/2012 and A/HongKong/2014) that were isolated between the first baseline and last follow-up as "recent strains". Strains isolated prior to an individual's birth were defined as "pre-birth strains" (Fig 1A–1C) and "post-birth strains" otherwise.

All individuals had detectable titers (1:10) and 95.6% of participants had titers of at least 1:40 to at least one strain at baseline (Fig 2A and 2B). Across all $21 \times 777$ strain-individual combinations, the most commonly detected titer value was < 1:10 at baseline (28.0%) and 1:20 and 1:40 at follow-up (22.0% and 21.8%, respectively) (Fig 2D and 2E). The median of geometric mean titer (GMT) across post-birth strains was 14.3 [interquartile range (IQR), 12.2 to 30.0) at baseline and 36.4 (IQR, 21.9 to 46.8) at follow-up (Fig 2A and 2B, and S2 Table). GMTs of post-birth strains [18.3; 95% confidence interval (CI), 18.0 to 18.7] were higher than that of pre-birth strains (9.6; 95% CI, 9.2 to 10.0) at baseline (Fig 2A and S2 Table). 73.9% of titers to pre-birth strains were below 1:40 at baseline (62.6% at follow-up; Fig 2A and 2B and 2G and 2H) and the lower titers were found to these pre-birth strains with the longer intervals from isolation to birth (S1 Fig). We fitted generalized additive models (GAM) to titers on age at isolation (i.e. age when the strain was isolated) and found the highest titers at a median age at isolation of 4.3 years (IQR, 2.0 to 6.9 years) across strains (Fig 3D and 3E; details in S1 Text). GMTs of recent strains (12.0; 95% CI, 11.6 to 12.4) were lower than that of non-recent strains

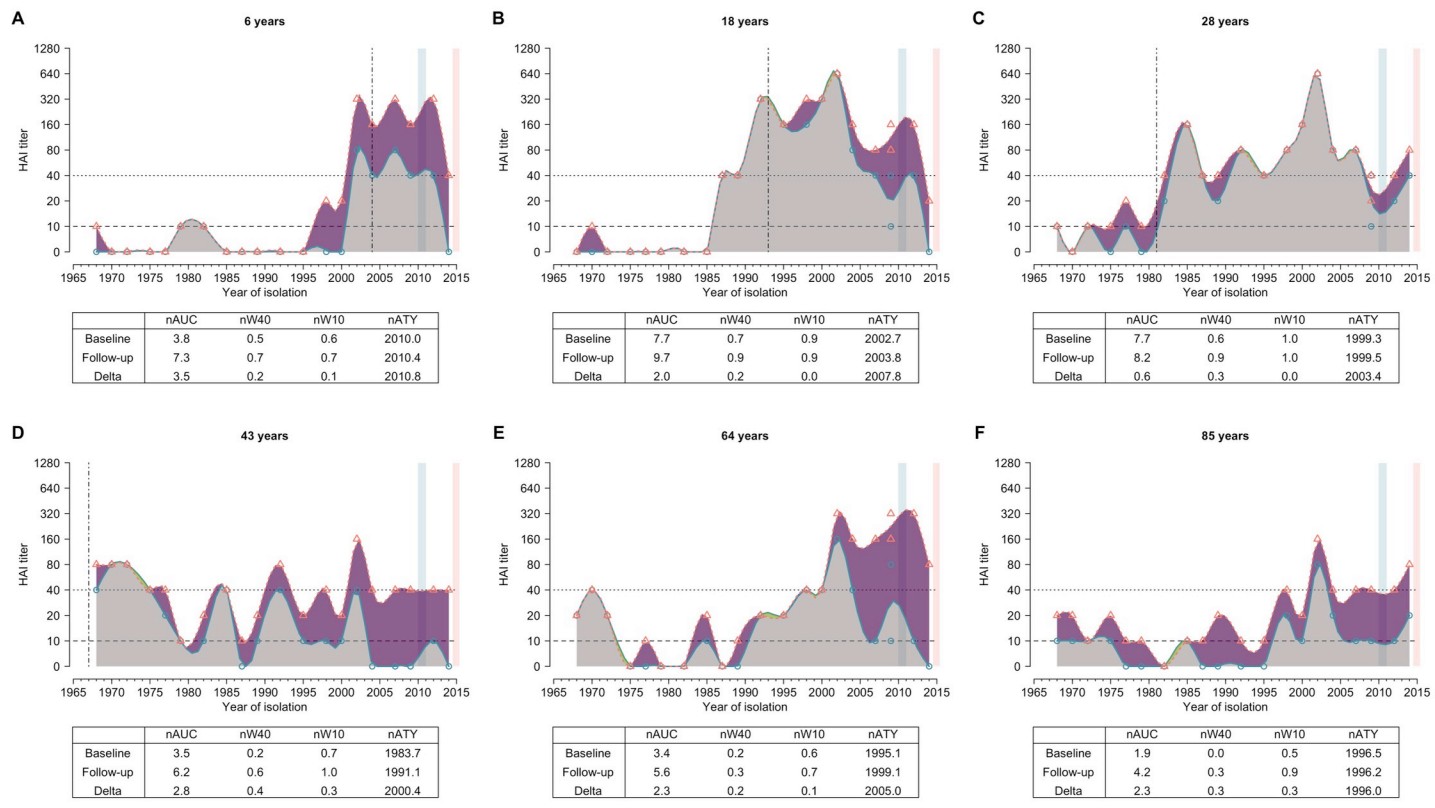

**Fig 1. Representative individual profiles of HAI titer against H3N2 strains circulating over forty years.** (A-F): Antibody profile for each representative individual. Blue circles and red triangles represent the HAI titers against the tested strains at baseline and follow-up visit, respectively. Blue and red solid lines represent the smoothed HAI titers for serum collected from baseline and follow-up visit, respectively. Smooth splines of HAI titers on circulating years are shown in this figure for illustration purposes and not used in the subsequent analysis. Grey areas represent the baseline antibody profile. Purple and green areas indicate the increase and decrease of HAI titer at follow-up visit compared to baseline, respectively. Blue and red vertical blocks represent the duration for baseline and follow-up visit, respectively. Vertical dotted-dashed lines indicate the year of birth of the individual. Dashed and dotted lines represent the titer of 1:10 (detectable cutoff) and 1:40 (protective cutoff), respectively. Table below each panel shows the values of metrics calculated for each individual. We provided six additional profiles of HAI titers from people who were aged 40 to 60 years in S1 Fig.

(18.7; 95% CI, 18.3 to 19.0) at baseline, but were higher at follow-up [41.5 (95% CI, 39.7 to 43.3) for recent strains and 31.3 (95% CI, 30.7 to 31.9) for non-recent strains].

## Summary metrics of antibody profiles

We hypothesized that features of an antibody profile determine an individual's odds of seroconversion over and above homologous titer. We developed several metrics aimed at summarizing the information in individuals', often complex, antibody profiles (Fig 4). We estimated: the *area under the curve* (AUC) for each antibody profile (i.e. the integral of an individual's measured log titers); the *width* ($W_Z$) of an individual's antibody titer above a threshold $z$ (i.e. the proportion of the profile above that threshold; $W_{40}$ for protective threshold and $W_{10}$ for detectable threshold); and the *average titer year* (ATY) of each antibody profile (i.e. the average of strain isolation years weighted by their titer) (see Methods). We hypothesized that these features of antibody profiles captured biologically relevant properties of the immune response to H3N2; in particular, overall levels of antibody mediated immunity (for AUC), the breadth of antibody mediated immune response (for $W_{40}$ and $W_{10}$) and temporal center of mass of H3N2 immunity (for ATY). In most analyses, we use normalized versions of these metrics (i.e. nAUC, $nW_{40}$, $nW_{10}$, nATY) to adjust for differences between individuals in the number of

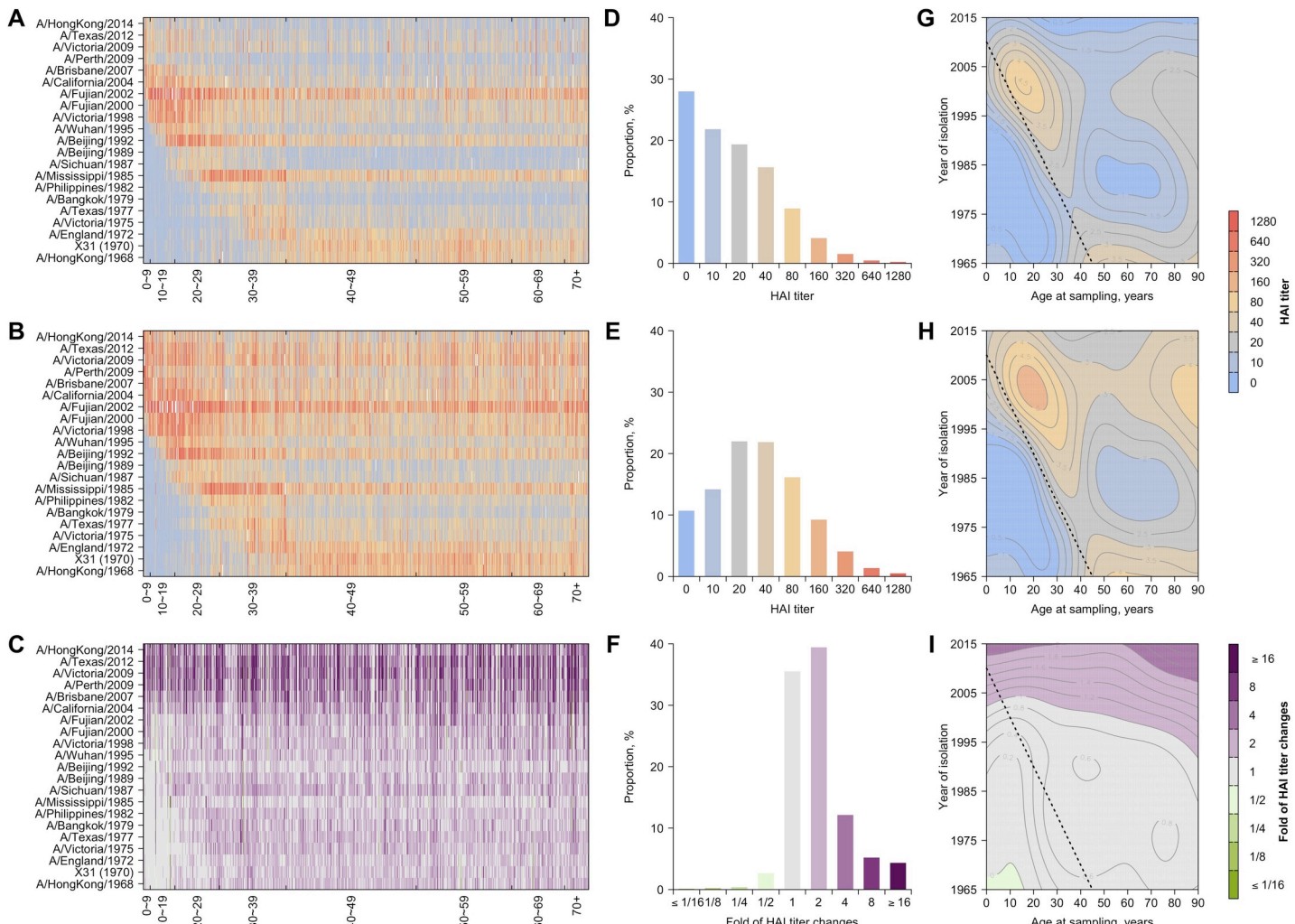

**Fig 2. HAI titers and differences in HAI titers between two visits against historical H3N2 strains.** (A) and (B): HAI titers against H3N2 strains from serum collected from baseline and follow-up visit, respectively. Participants are sorted by age at baseline sampling. (C): Fold of changes of HAI titers between two visits. Cell of row *i* and column *j* represents HAI titer or differences of HAI between two visits to strain *i* for person *j*. (D-F): Distribution of HAI titers from serum collected from baseline (D), follow-up visit (E), and fold of changes between two visits (F), respectively. Distributions were plotted after grouping titers across all participants and all tested H3N2 strains. (G) Variations of HAI titers at baseline with ages of participants and the year of isolation of tested strains. Dashed lines represent birth years of the participants (same for panel H and I). (H) Variations of HAI titers at follow-up with ages of participants and the year of isolation of tested strains. (I) Variation of changes in HAI titers between the two visits with ages of participants and the year of isolation of tested strains.

possibly exposed strains given their ages (i.e. individuals could not have been exposed to pre-birth strains) (see Methods. Non-normalized analysis included in S1 Text, S2 Fig, S3 and S4 Tables).

We calculated nAUC, nW$_{40}$, nW$_{10}$, nATY for each participant, and fitted generalized additive models using a spline on age to estimate the association of these metrics with participant age. The nAUC of titers increased with participant age, peaking at ~ 20 years of age for both visits and gradually decreasing to a low at ~ 50 years of age (Fig 4B and 4C). The average nAUC among participants aged 50 years old was estimated to be lower than that among participants aged 20 years old [ratio: 0.48 (95% CI, 0.42 to 0.53; 95% CI from 1000 bootstraps and see Methods for details) for baseline and 0.59 (95% CI, 0.53 to 0.67) for follow-up]. After the age of 70, nAUC started to increase.

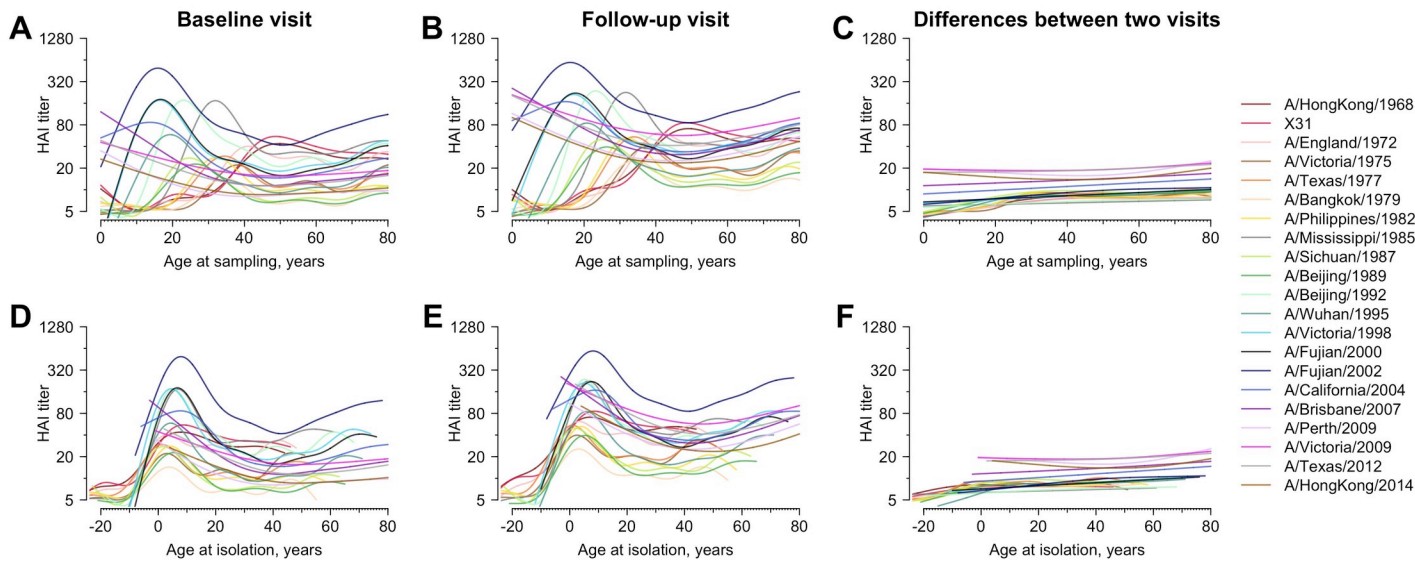

**Fig 3. Age and HAI titer against individual H3N2 strain.** Lines are the predicted mean HAI titer fitted from general additive model (GAM) using age at sampling (panel A to C) and age at isolation (panel D to F) as predictor, respectively. We fitted separate GAMs for HAI titers measured for serum collected from baseline (panel A and D), serum collected in follow-up visit (panel B and E) and the differences between two visits (panel C and F).

The width of HAI titers above protective titer levels ($nW_{40}$) illustrated an increasing trend with age grows until approximately 15 years of age at the time of sample collection, peaking at 60.3% and 80.1% at baseline and follow-up, respectively (Fig 4F and 4G). There was a more distinct drop in $nW_{40}$ among participants aged 50 years old [ratio of $nW_{40}$ at 50 to $nW_{40}$ at 15: 0.28 (95% CI, 0.25 to 0.29; 95% CI from 1000 bootstraps and see Methods for details) and 0.46 (95% CI, 0.43 to 0.49) for baseline and follow-up, respectively], compared to the drop in $nW_{10}$ [ratio of $nW_{10}$ at 50 to $nW_{10}$ at 15: 0.78 (95% CI, 0.75 to 0.79) and 0.94 (95% CI, 0.90 to 0.97)] (S3 Fig). An uptick of widths among those older than 60 years was observed for both cutoffs (Figs 4 and S3).

The intent of nATY is to help us understand how immunity to strains circulating at different times contributes to the overall immune profile. At the extremes, if the antibody response was completely dominated by early infections, nATY would track with birth year. However, we hypothesized that people constantly generate updated antibody responses to newly encountered strains during their lifetimes, and nATY would track with the midpoint of the post-birth strains (i.e. unweighted average isolation year of post-birth strains). Our empirical observations are in line with our hypothesis; baseline nATY moved away from birth year at a rate of 0.50 years for every additional year of life (0.45 years at follow-up) and tracked the midpoint of the post-birth strains (Fig 4J and 4K). The average nATY among people aged 40 years or older, who have been exposed to all tested strains, stayed unchanged and centered on the unweighted average isolation year of all tested strains. The follow-up nATY tended to skew 2.4 years to more recent years than the unweighted average isolation year among those people who seroconverted to the recent strains (t-test, p < 0.01).

## Changes in antibody profiles between the two visits

Across our cohort, changes, particularly increases, in titer were seen across all strains, including strains that have long been extinct (Figs 2C–2F and 5A). 97.9% of people experienced a rise against one or more strains. 73.7% showed a 4-fold or greater titer increase (seroconversion) to one or more, with increased risk of occurring in strains circulating after 1998 compared to A/

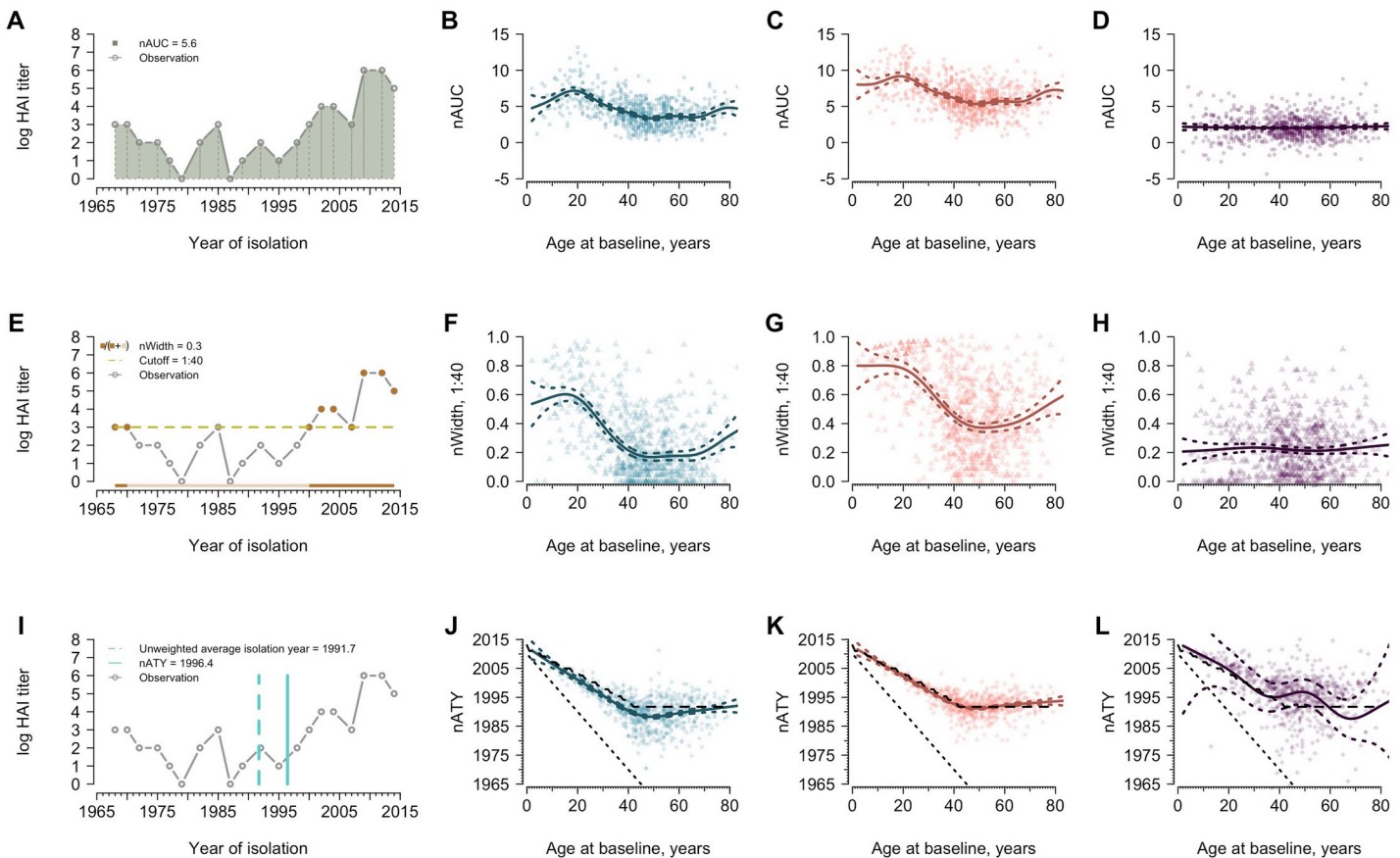

**Fig 4. The normalized area under the curve (nAUC), width above 1:40 (nW$_{40}$) and average titer years (nATY) varying with age.** Metrics were calculated using post-birth strains and normalized by the number of post-birth strains. Blue and red represent the metrics measured for serum collected from baseline and follow-up visit, respectively. Purple indicates the differences in metrics between the two visits. Solid lines are predictions from a generalized additive model and the colored dashed lines represent the corresponding 95% confidence intervals. (A) Demonstration of nAUC for one participant as an example. The same participant, who was aged 73 years old at baseline, is used for panel E and I. (B) Age and nAUC at baseline. (C) Age and nAUC at follow-up. (D) Age and changes in nAUC between the two visits. (E) Demonstration of nW$_{40}$ for one participant as an example. Solid points are the years that contributed to calculating width. (F) Age and nW$_{40}$ at baseline. (G) Age and nW$_{40}$ at follow-up. (H) Age and changes in nW$_{40}$ between the visits. (I) Demonstration of nATY for one participant as an example. (J) Age and nATY at baseline. The sloping black dotted lines indicate the year of birth of participants and the black dashed lines indicate the unweighted average isolation year of post-birth strains given age on x-axis (same for panel K and L). (K) Age and nATY at follow-up. (L) Age and changes in nATY between the visits.

HongKong/1968 [odds ratio (OR) ranges from 2.7 (95% CI, 2.0 to 3.9) to 16.4 (95% CI, 12.0 to 22.6) across these strains] (S4 Fig), i.e. strains more antigenically similar to ones with which the participant could have been infected. The largest increases in HAI titers were detected for four recent strains (Fig 5B), and there was high-correlation in seroconversions; 63.1% of those who seroconverted to one or more strains seroconverted to all four strains. (S5 Fig). Meanwhile, a minority of people (18.3%) showed a decrease in titer to one or more strains, likely due to transient boosting from exposures prior to baseline sampling [5,21].

## Effect of pre-existing immunity on seroconversion to recent strains

Pre-existing, homologous titer to circulating strains is a well-described predictor of odds of seroconversion [14], which is confirmed in our study (Tables 1, S3 and S4). However, we hypothesized that including metrics of antibody profiles that integrate previous exposure

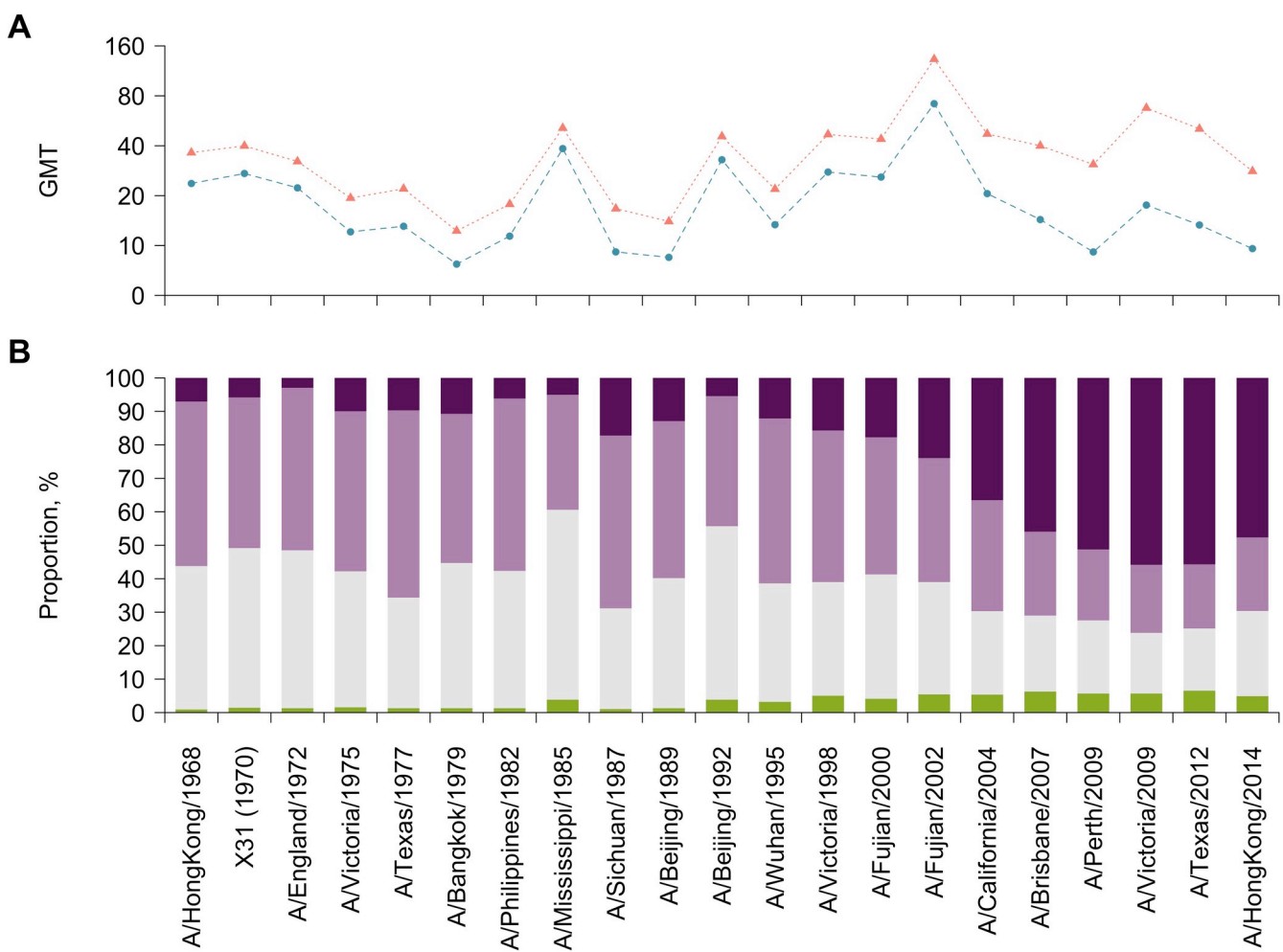

**Fig 5. Changes in antibody profiles between visits.** (A) Geometric mean titers (GMT) by strain. Blue and red points represent GMT at baseline and follow-up, respectively. (B) Distribution of changes in titers by strain. We divided the changes in titers into four categories, i.e. decrease (green), no change (grey), two-fold increase (light purple) and four-fold change (seroconversion, dark purple).

history would further improve predictions of the odds of seroconversion. Therefore, we examined the association between the odds of seroconversion with age at baseline sampling, pre-existing, homologous titer to outcome strains (the *i-1*th strain), titer to the *i-1*th strain, and summary metrics (i.e. nAUC, $nW_{10}$, $nW_{40}$, and nATY) of strains up to and including the *i-2*th strain ("immunity of non-recent strains" hereafter). A greater frequency of seroconversion to an examined strain is expected when the estimated odds ratio (OR) of the above-mentioned predictors is greater than 1 (Table 1). We found including nAUC or $nW_{40}$ improved the deviance explained (e.g. 11% increase in deviance explained after including nAUC for A/Texas/2012; Models 1–2 in Table 1) and Akaike information criterion (AIC) (S6 Fig) of the models. After adjustment for pre-existing titer to strain *i*, we found pre-existing nAUC and $nW_{40}$ were each positively associated with seroconversion (Tables 1, S3 and S5). Taking A/Texas/2012 as an example, pre-existing titer to A/Texas/2012 was negatively associated with the odds of seroconversion (OR: 0.45; 95% CI, 0.35 to 0.57; Model 3 in Table 1), while there was an increased odds of seroconversion associated with higher pre-existing nAUC (OR: 1.21; 95% CI, 1.10 to 1.34) and $nW_{40}$ (OR: 5.53; 95% CI, 2.15 to 14.57).

**Table 1. Associations between pre-existing immunity and seroconversion to four recent strains.**

| | Adjusted odds ratio (95% confidence interval) | | | |
|---|---|---|---|---|
| | A/Perth/2009 | A/Victoria/2009 | A/Texas/2012 | A/HongKong/2014 |
| **Model 1** | | | | |
| Age at sampling | 1.00 (0.99, 1.01) | 0.99 (0.98, 1.00) | 0.99 (0.98, 1.00)* | 1.00 (0.99, 1.01) |
| Titer to strain $i$ [a] | 0.43 (0.34, 0.54)* | 0.51 (0.42, 0.60)* | 0.49 (0.38, 0.61)* | 0.70 (0.56, 0.85)* |
| Titer to strain $i-1$ [a] | 1.30 (1.10, 1.55)* | 1.03 (0.87, 1.22) | 1.06 (0.85, 1.32) | 0.93 (0.79, 1.09) |
| Deviance explained | 7.5% | 12.9% | 12.6% | 4.0% |
| **Model 2** | | | | |
| Age at sampling | 1.01 (1.00, 1.02) | 1.00 (0.99, 1.01) | 1.00 (0.99, 1.01) | 1.00 (0.99, 1.01) |
| Titer to strain $i$ | 0.42 (0.33, 0.53)* | 0.48 (0.40, 0.57)* | 0.45 (0.35, 0.57)* | 0.67 (0.54, 0.82)* |
| Titer to strain $i-1$ | 1.21 (1.02, 1.44)* | 0.97 (0.82, 1.16) | 0.97 (0.78, 1.22) | 0.85 (0.71, 1.01) |
| AUC [b] | 1.14 (1.05, 1.24)* | 1.17 (1.07, 1.29)* | 1.21 (1.10, 1.34)* | 1.17 (1.07, 1.29)* |
| Deviance explained | 8.3% | 14.0% | 14.0% | 5.0% |
| **Model 3** | | | | |
| Age at sampling | 1.00 (0.99, 1.02) | 1.00 (0.99, 1.01) | 1.00 (0.99, 1.01) | 1.00 (0.99, 1.01) |
| Titer to strain $i$ | 0.43 (0.34, 0.54)* | 0.49 (0.40, 0.58)* | 0.45 (0.35, 0.57)* | 0.68 (0.55, 0.84)* |
| Titer to strain $i-1$ | 1.24 (1.04, 1.47)* | 0.98 (0.83, 1.17) | 1.00 (0.80, 1.25) | 0.88 (0.74, 1.05) |
| Width, cut off 1:40 [b] | 2.86 (1.20, 6.92)* | 4.50 (1.77, 11.71)* | 5.53 (2.15, 14.57)* | 2.79 (1.13, 6.98)* |
| Deviance explained | 8.0% | 13.9% | 13.8% | 4.5% |

[a] Strain $i$ refers to the strain that was examined for seroconversion, and strain $i-1$ refers to the most recent strain isolated prior to strain $i$. E.g. when using seroconversion to A/Perth/2009 as outcome, strain $i$ and $i-1$ will be A/Perth/2009 and A/Brisbane/2007, respectively.

[b] Metrics were calculated using titers to strains isolated after the birth of the participants and before the year that strain $i-1$ was isolated. Adjustment was then performed by standardizing the metrics with the number of post-birth strains.

* Statistical significant level of 0.05.

We did not observe a positive association between immunity to non-recent strains and seroconversion to recent strains in the univariable analysis (S6 Table), which was expected given that people with higher immunity of non-recent strains tended to have higher homologous titers to recent strains (S7 Table). We therefore performed mediation analysis under the hypothesis (S7 Fig) that pre-existing immunity to pre-exposures imposes both indirect effect mediated by titer to strain $i$ and direct effect on seroconversion to strain $i$. Results suggested both AUC and $nW_{40}$ had negative indirect effects and positive direct effects (except for $nW_{40}$ on A/HongKong/2014) on seroconversion to strain $i$, which yielded non-significant or marginally negative total effects (S8 Fig).

## Discussion

### Life course exposures continually shape antibody profiles of influenza

We found an individual's antibody profile is dynamically updated across time. A majority of participants experienced seroconversion to recent strains and a minority of participants showed decreases in antibody profiles due to the transient antibody dynamics. Therefore, we expected that titers to recent strains would vary over the course of our study, but that, whereas titers to non-recent strains would to be relatively constant and when variation did exist, it would be in a random direction. However, we found that antibody titers to non-recent strains overwhelmingly increased (Figs 2C and 4B), inconsistent with random variation (S9 Fig) suggesting that most increases, even those to non-recent strains, can be attributed to exposure to circulating strains. Of note, the HI titers and rises in titers to A/HongKong/2014 were relatively lower compared to other recent strains, which may due to low assay sensitivity to detect HI titers

to Clade 3c.2a strains and/or the co-circulation with the Clade 3c.3a strains (e.g. A/Switzerland/ 2013). Though titer to non-recent strains in general increased between the two samples, the shape of each person's antibody profile was highly consistent between visits (median Pearson's correlation 0.90; IQR, 0.82 to 0.94). This result is similar to previously described results from 5 landscapes for 69 individuals [7], suggesting that individuals maintain patterns of relative titer across historical strains, even while acquiring antibodies due to new exposures.

We expect individuals of different ages to show different antibody profiles due to differences in lifetime influenza exposures. However, we found that our summary metrics did not monotonically increase with age. There is a gradual increase in the overall (nAUC) and breadth of immunity ($nW_{40}$) throughout childhood consistent with the accumulation of exposures (Figs 3, 4J and 4K) [3–5]. The observed decrease in nAUC and $nW_{40}$ among people aged 40 to 50 and age-independent changes in these metrics (Fig 4D and 4H) could suggest low HAI antibody among this age group, which could be due to high non-specific immune responses [e.g. antibodies against HA stalk and neuraminidase (NA) and cellular immune responses; [21–25]]. These non-specific immunities, which may accumulate over multiple exposures and are not measured in our assay, could prevent people from being infected and producing updated, strain specific HA responses [22,23,26].

Our hypotheses about the mechanisms driving age patterns in antibody profiles implicitly assume repeated, regular exposure to influenza throughout life. Consistent with this assumption, our findings during this four-year period suggested a nearly universal exposure to H3N2 in this cohort (Figs 2C and 4D), agreeing with observations of high H3N2 incidence during this period with particularly large numbers of cases in 2011–12 that was covered by our study period [19,20]. This is consistent with other reports that the H1N1 pandemic which occurred just before or just after the beginning of our baseline sampling lead to a period of low H3N2 incidence, followed by a large resurgence of H3N2 in the period 2011–2012 [27]. 24.1% of our participants only had two-fold increase to the tested strains, suggesting widely existing exposures but probably not infections.

## Future immune responses to influenza are driven by antibody profiles

Our work further quantified the complex role that antibody profiles play in future immune responses to influenza. We confirmed the prominent role of pre-existing, homologous titers in determining the odds of seroconversion. However, we found that participants who had higher immunity to previously exposed strains were more likely to experience seroconversion to recent strains after adjusting for homologous titer. These findings were not affected by age, self-reported vaccination or collection time (S10 and S11 Figs and S5, S8–S10 Tables). Given individuals with high immunity to non-recent strains tended to have higher homologous titers (S6 and S7 Tables), we investigated whether the effect of immunity to non-recent strains was mediated by homologous titer (S7 and S8 Figs; detailed in S1 Text) and found that pre-existing immunity to non-recent strains imposes a positive direct effect on the odds of seroconversion to recent strains but a negative, homologous titer mediated, indirect effect.

The mechanism behind the positive association between immunity to non-recent strains and seroconversion to recent strains is unclear. One plausible hypothesis is that a subset of people tend to have a more vigorous titer response across strains (e.g. individual heterogeneity in immune responses). This is supported by our data that positive association can still be detected when using titer to an older strain, instead of the summary metric of antibody profiles, as the proxy of immunity to non-recent strains in models shown in Table 1 (S12 Fig). Another plausible hypothesis is that non-HAI immunity that was acquired from previous infections (e.g. antibody to HA stalk and NA and cellular immune responses) could blunt the

production of strain-specific antibody upon exposure to circulating strains [21–24], thus reducing the HAI titers to the circulating strains and increasing the probability of seroconversion. High antibody to non-recent strains could indicate individuals who have not experienced infection in recent times [e.g. low strain-specific antibody to currently circulating strains due to temporary protection from last infection [26,28,29]]. Homologous HAI titers may reflect the combined binding ability of antibody targeting the tested antigen as well as antibody targeting previously encountered epitopes, and high antibody titers to non-recent strains may indicate particular distributions of these two that are less protective.

### Life course immunity of influenza provides new opportunities in influenza studies

Previous work on mapping antigenic distance of evolving strains and characterization of life course immunity has changed how we approach studies of immunity to influenza [2,7]. The summary metrics we developed allow efficient characterization of profiles that are amenable to broad use in the analysis. In particular, although the age patterns in AUC and widths are similar, width is more likely to capture specific responses to a HA and therefore the two metrics would diverge in a person who does not have strong responses to a strain and its antigenically relatives. With these metrics, we integrated information of life course HAI antibody mediated immunity to influenza that accounted for complex exposure histories and cross-reactions between influenza strains. These metrics should help to quantify the life course of non-HAI immunity that were not measured by our assay, and the interactions between infection events and immunity to influenza through life.

### Limitations

Our study has several limitations. First, we didn't include strain from Clade 2c.3a, which was co-circulating with A/HongKong/2014-like strain in the 2014–2015 season [30]. Our results seem to be robust as we found similar effects of immunity to previous exposures on the seroconversion to four recent strains (Table 1). Second, between-subtype interactions have not been incorporated into the work, especially interactions with H1N1 which might confer temporary protection against H3N2 [31]. We, however, observed inconsistent responses to H3N2 strains that were isolated after the 1977 epidemic and 2009 pandemic of H1N1 (Fig 2B and 2H).

### Conclusion

We developed multiple summary metrics to integrate information of antibody profiles, which enabled us to further demonstrate clear age patterns of these profiles. We found accumulation of immunity to H3N2 during the first twenty years of life, followed by reductions among 40–50 years old. These age patterns in antibody profiles are likely to be driven by continual exposure to influenza; our study suggested nearly universal exposures during a four-year interval. Meanwhile, antibody profiles were found to provide more information than homologous titers in predicting temporal changes in influenza immunity. In particular, higher immunity to older strains was found associated with increased odds of seroconversion to the currently circulating strains, which mechanism remains unclear.

## Materials and methods

### Ethics statements

Study protocols and instruments were approved by the following institutional review boards: Johns Hopkins Bloomberg School of Public Health, University of Florida, University of

Liverpool, University of Hong Kong, Guangzhou No. 12 Hospital, and Shantou University. Written informed consent was obtained from all participants over 12 years old; verbal assent was obtained from participants 12 years old or younger. Written permission of a legally authorized representative was obtained for all participants under 18 years old.

## Cohort profile

The Fluscape study is an ongoing cohort study in and around Guangzhou City in southern China that collects sera from consenting participants at regular intervals (roughly annually), as well as information on demographics. The methods and study population have been described in detail elsewhere [18]. Briefly, 40 locations were randomly selected from a spatial transect extending to the northeast from the center of Guangzhou. In each location 20 houses were randomly selected and all consenting residents over 2 years of age were enrolled in the study. When households left the study, they were replaced by new randomly selected households to maintain a population of 20 households per location.

We collected serum from participants at two time points, i.e. baseline (2009 to 2011) and follow-up (2014–2015). Serum from both visits were available from 777 participants out of 2,767 total participants (across both visits, S1 Table), of which 763 participants had interpretable titers for all 21 H3N2 strains (Fig 2A–2C, 14 participants had uninterpretable results for 5 strains). Age at the time of the baseline ranged from 2 to 86 years old, with over representation of participants aged 40–59 years compared to the Chinese population according to the 2009 census (51.4% vs. 31.6%) [18].

## Laboratory testing

Blood samples were kept at 4˚C until processing on the day of collection. Serum is extracted from these samples and frozen at -80˚C until testing. For all individuals recruited at both baseline visit (4 December 2009 to 22 January 2011) and follow-up visit (17 June 2014 to 2 June 2015) we measured hemagglutination inhibition (HAI) titers for antibodies against a panel of 21 strains of H3N2 influenza spanning the history of the virus since its emergence in humans in 1968 (A/Hong Kong/1968, X31, A/England/1972, A/Victoria/1975, A/Texas/1977, A/Bangkok/1979, A/Philippines/1982, A/Mississippi/1985, A/Sichuan/1987, A/Beijing/1989, A/Beijing/1992, A/Wuhan/1995, A/Victoria/1998, A/Fujian/2000, A/Fujian/2002, A/California/2004, A/Brisbane/2007, A/Perth/2009, A/Victoria/2009, A/Texas/2012, A/Hong Kong/2014). The 50% tissue culture infectious dose (TCID50) for each virus was determined on Madin-Darby canine kidney (MDCK) cells [32]. For each strain-individual pair, HAI titers were determined by two-fold serial dilutions from 1:10 to 1:1280 conducted in 96-well microtiter plates with 0.5% turkey erythrocytes. Sera from the two visits were tested side by side on the same plate, and confirmatory samples collected in 2014 sera were tested on a separate plate. The reciprocal of the highest dilution where hemagglutination does not occur is reported as the titer. There were 28 missing (for 14 individuals and 5 strains) out of 32,636 strain-individual-visit combinations, which were due to inadequate sera remaining or inconclusive readings.

## Analytic methods

For analysis, individuals with undetected titers are assumed to have a titer of 5, and all titers are transformed to the log-scale. Specifically, we transform titers based on the formula $y = log_2(x/5)$ where x is the measured titer, which results in a 0 for undetectable titers and a 1 unit rise in titer for each 2-fold increase (so 10 = 1, 20 = 2, 40 = 3, etc.). In figures, measures are transformed back to the arithmetic scale for clarity; but all statistics are presented on this scale. Geometric mean titer (GMT) was calculated as the mean of log-titer and transformed back to the

arithmetic scale. 95% confidence intervals (CIs) were calculated using student t tests assuming the log-titer follows a normal distribution. Age at baseline sampling is defined as the time difference between the year of baseline and the year of birth; age at isolation is calculated as the year when the tested strain was isolated minus the year of birth of the participant.

**Age patterns in strain-specific titers.** We fitted univariable GAMs to the measured log-titers on the participants' age at baseline sampling (i.e. participant age when serum was collected at baseline sampling) and age at isolation [i.e. participants' ages when strains were isolated [4]], respectively to examine the strain-independent age patterns of HAI titers (Fig 3). We fitted separate models to each strain. From each fitted model, we derived the age at sampling or age at isolation when the titer to each strain is predicted to be the highest and characterized the range across strains.

**Summary metrics.** Individual antibody profiles were constructed using each individual's HAI titers against 21 tested H3N2 strains which were sorted in chronological order. We characterized the shape of these individual antibody profiles using a number of statistics (Fig 4) which are proposed to approximately estimate the area under the antigenic landscape surface (Area Under the Curve, AUC), the breadth of antibody profile (Width, W), where the breadth is calculated above either detective (1:10) or protective threshold (1:40), and the temporal targeting of antibody profile by the measured HAI titer values (Averaged Titer Year, ATY). The statistics are calculated as follows:

1. *Area Under the Curve (*AUC*)*: We estimated the area under the curve of antibody profile as:

$$AUC_{j,v} = \sum_{i=1}^{M-1} \frac{y_{i,j,v} + y_{i+1,j,v}}{2} \left( t_{i+1} - t_i \right)$$

where $AUC_{j,v}$ is the area-under the curve of titers by time for person $j$ and visit $v$, $M$ is the total number of included strains, $y_{i,j,v}$ is participant $j$'s log-titer against strain $i$ at visit $v$, and $t_i$ is the time of isolation of strain $i$.

2. *Width (*W*)*: We define the width of the curve to be the proportion of time during which the antibody profile that is greater than or equal to some predefined antibody titer cutoff, Z. Here we focus on detectable titers ($W_{10}$, Z = 1:10), and protective titers ($W_{40}$), based on the commonly used cutoff Z = 1:40. When performing the calculation, we transformed the threshold to log-scale based on the formula z = log2(Z/5). Hence,

$$W_{Z,j,v} = \sum_{i=1}^{M-1} W_{Z,j,v}(t_i, t_i + 1)$$

where:

$$W_{Z,j,v}\left(t_i, t_{i+1}\right) = \begin{cases} t_{i+1} - t_i, & y_{i+1,j,v} \geq z \text{ and } y_{i,j,v} \geq z \\ 0, & y_{i+1,j,v} < z \text{ and } y_{i,j,v} < z \\ \dfrac{(y_{i+1,j,v} - z)(t_{i+1} - t_i)}{y_{i+1,j,v} - y_{i,j,v}}, & y_{i+1,j,v} \geq z \text{ and } y_{i,j,v} < z \\ \dfrac{(z - y_{i,j,v})(t_{i+1} - t_i)}{y_{i+1,j,v} - y_{i,j,v}}, & y_{i+1,j,v} < z \text{ and } y_{i,j,v} \geq z \end{cases}$$

When titers to all strains are above the threshold $z$, the width for an individual given the

tested strains is at its maximum:

$$max(W_{Z,j,v}) = \sum_{i=1}^{M-1} (t_{i+1} - t_i) = t_M - t_1$$

We present the width (ranges from 0 to 1) standardized by the maximum width in the results:

$$sW_{Z,j,v} = \frac{1}{t_M - t_1} W_{Z,j,v}$$

3. *Average Titer Year (ATY)*: The average titer year is the center of mass of the curve with respect to strain isolation time, capturing all of an individual's titer values (i.e., the weighted average of their titers):

$$ATY_{j,v} = \frac{1}{AUC_{j,v}} \sum_{i=1}^{M-1} \frac{(y_{i,j,v} + y_{i+1,j,v})}{2} \frac{(t_{i+1} + t_i)}{2} (t_{i+1} - t_i)$$

In order to account for the impact of pre-birth strains, we calculated the metrics using post-birth strains and normalized with the number of post-birth strains ($M_{post}$) in the main analysis. Exclusion of pre-birth strains was performed for each participant according to the year of birth. The normalized statistics are calculated as follows:

1. *Normalized Area Under the Curve (nAUC)*:

$$nAUC_{j,v} = \frac{1}{M_{post}} \sum_{i=1}^{M_{post}-1} \frac{y_{i,j,v} + y_{i+1,j,v}}{2} (t_{i+1} - t_i)$$

2. *Normalized Width (nW)*:

$$nW_{Z,j,v} = \frac{1}{t_{M_{post}} - t_1} \sum_{i=1}^{M_{post}-1} W_{Z,j,v}(t_i, t_{i+1})$$

3. *Normalized Average Titer Year (nATY)*:

$$nATY_{j,v} = \frac{1}{M_{post} nAUC_{j,v}} \sum_{i=1}^{M_{post}-1} \frac{(y_{i,j,v} + y_{i+1,j,v})}{2} \frac{(t_{i+1} + t_i)}{2} (t_{i+1} - t_i)$$

For both normalized and unnormalized metrics, the changes in area under the curve and width between the visits were calculated as the difference between baseline and follow-up values of these metrics. In order to examine the temporal targeting of the changes in titers between the two visits, the changes in nATY (Fig 4L) and ATY (S2 Fig) are calculated using

difference in titers:

$$\Delta nATY_j = \frac{1}{\sum_{i=1}^{M_{post}-1} \frac{\Delta y_{i,j}+\Delta y_{i+1,j}}{2}\left(t_{i+1}-t_i\right)} \sum_{i=1}^{M_{post}-1} \frac{\left(\Delta y_{i,j}+\Delta y_{i+1,j}\right)}{2}\frac{\left(t_{i+1}+t_i\right)}{2}\left(t_{i+1}-t_i\right)$$

$$\Delta ATY_j = \frac{1}{\sum_{i=1}^{M-1} \frac{\Delta y_{i,j}+\Delta y_{i+1,j}}{2}\left(t_{i+1}-t_i\right)} \sum_{i=1}^{M-1} \frac{\left(\Delta y_{i,j}+\Delta y_{i+1,j}\right)}{2}\frac{\left(t_{i+1}+t_i\right)}{2}\left(t_{i+1}-t_i\right)$$

**Age pattern in antibody profiles.** To account for the heterogeneous exposure history of participants at different ages (Fig 2G–2I), we fitted generalized additive models (GAM) to the measured log-titers ($y_{i,j}$) against all tested H3N2 strains incorporating a smoothed interaction term of age at sampling ($a_{j,1}$) and year of the strains circulated ($t_i$):

$$y_{i,j} = s(t_i, a_{j,1})$$

We also fitted separate GAM to examine the non-linear association between each of summary analytic statistics ($I_{j,v} \in \{AUC_{j,v}, ATY_{j,v}, W_{z,j,v}\}$) mentioned in the above section and the ages of participants at baseline sampling (Figs 4, S2 and S3) with the equation below:

$$I_{j,v} = s(a_{j,1})$$

We used the ratio of the average nAUC among people aged 50 years old to that among people aged 20 years old, to characterize the relative reduction of nAUC among people at 50 to the peak value. We derived the 95% CI of the ratio through 1000 bootstraps. In each bootstrap, we resampled the data set of observed nAUC and age at sampling, and refitted the above-mentioned GAM. We then calculated the ratio from each model prediction. With the same method, we also calculated the ratio of the average nW among people aged 50 years old to that among people aged 15 years old, at which the average nW is the greatest.

**Changes in antibody profiles between visits.** The change of HAI titer is defined as the difference in the measured log-titers against tested H3N2 strain between the baseline and follow-up visits. We characterized the strain distribution among titers that were decreased, no change, two-fold increase and four-fold increased (seroconversion), respectively and compared such distribution with the underlying strain distribution (i.e. the number of titers of that strain divided the number of titers of all tested strains) (S5 Fig). We fitted logistic regression of seroconversion ($c_{i,j}$) on strains ($s_i$, *as a categorical variable*) and adjusted for age at baseline sampling ($a_{j,1}$) and prior log-titers ($y_{i,j,1}$), to further examine the strain-specific odds of seroconversion ($\alpha_3$, S4 Fig):

$$logit\left(p(c_{i,j}=1)\right) = \alpha_0 + \alpha_1 a_{j,1} + \alpha_2 y_{i,j,1} + \alpha_3 s_i$$

We characterized the changes of HAI titers to four recent strains that were possibly circulating during our study period, i.e. A/HongKong/2014, A/Texas/2012, A/Victoria/2009 and A/Perth/2009. Distribution of changes by the number of strains that participants showed increased titers to and by individual strain was characterized, respectively (S5 Fig). 95% confidence intervals were calculated by assuming a binomial distribution.

**Effects of pre-existing immunity on seroconversion to recent strains.** We predicted the odds of seroconversion ($c_{i,j}$) to one of four recent strains $i$ (i.e. A/HongKong/2014, A/Texas/2012, A/Victoria/2009 or A/Perth/2009) by fitting logistic regression with predictors that reflect varying durations of exposure history and adjusting for age at baseline sampling ($a_{j,1}$). In brief, we progressively included log-titer to the examined strain (*i*th, $y_{i,j,1}$), log-titer to the

strain prior to the examined strain ($i$-$1$th, $y_{i-1,j,1}$) and one of the summary metrics ($I_{j,1} \in \{AUC_{j,1}, ATY_{j,1}, W_{z,j,1}\}$) that was calculated using strains isolated since the year of birth (or 1968 when including pre-birth strains) to the year before the $i$-$1$th strain was isolated. The model adjusting for the most complete exposure history is:

$$logit(p(c_{i,j} = 1)) = \gamma_0 + \gamma_1 a_{j,1} + \gamma_2 y_{i,j,1} + \gamma_3 y_{i-1,j,1} + \gamma_4 I_{j,1},$$

and the model was fitted to each recent strain separately. Akaike information criterion (AIC) and Bayesian information criterion (BIC) were used to compare the performance of the models (S6 and S13 Figs). We present results of three models that included different predictors in Table 1; only including log-titer to strain $i$ and strain $i$-$1$ (Model 1), including log-titer to strain $i$ and strain $i$-$1$ and $AUC_{j,1}$ (Model 2), and including log-titer to strain $i$ and strain $i$-$1$ and $W_{40,j,1}$ (Model 3). We considered a linear effect of age on seroconversion to recent strains after adjusting for immunity to previous exposures in the main analyses as we hypothesized the latter could account for the non-linear effect of age (Table 1 and S6 Fig). We also considered a non-linear effect of age on seroconversion to recent strains in the sensitivity analyses and found qualitatively similar results with respect to the effect of other covariates in models (S5 Table and S13 Fig).

In order to further separate the direct effect of immunity of non-recent strains from the indirect effect, we conducted a mediation analysis based on the causal diagram shown in S7 Fig. We first fitted the mediation model using a linear regression of pre-existing log-titer to strain $i$ ($y_{i,j,1}$) on pre-existing immunity to previous exposures ($I_{j,1}$) and adjusting for age at baseline ($a_{j,1}$). We then fitted the outcome model using a logistic regression of seroconversion to strain $i$ ($c_{i,j}$) on pre-existing log-titer to strain $i$ ($y_{i,j,1}$), pre-existing immunity to previous exposures ($I_{j,1}$) and age at baseline ($a_{j,1}$). We estimated total effect, average direct effect and average indirect effect using the "mediation" package with 1,000 bootstrap samples [33]. We also fitted the outcome model with an additional interaction term between pre-existing log-titer to strain $i$ and pre-existing immunity to previous exposures in order to account for the effect of their interactions on the mediation effects (S8 Fig).

All analyses were performed using R version 3.5.0 (R Foundation for Statistical Computing, Vienna, Austria), among which we fitted GAM using "mgcv" package [34].

## Supporting information

**S1 Text.**
(DOCX)

**S1 Fig. Representative individual profiles of HAI titer against H3N2 strains circulating over forty years among people aged 40 to 60 years.** (A-F): Antibody profile for each representative individual aged 40–60 years. Blue circles and red triangles represent the HAI titers against the tested strains at baseline and follow-up visit, respectively. Blue and red solid lines represent the smoothed HAI titers for serum collected from baseline and follow-up visit, respectively. Smooth splines of HAI titers on circulating years are shown in this figure for illustration purposes and not used in the subsequent analysis. Grey areas represent the baseline antibody profile. Purple and green areas indicate the increase and decrease of HAI titer at follow-up visit compared to baseline, respectively. Blue and red vertical blocks represent the duration for baseline and follow-up visit, respectively. Vertical dotted-dashed lines indicate the year of birth of the individual. Dashed and dotted lines represent the titer of 1:10 (detectable cutoff) and 1:40 (protective cutoff), respectively.
(TIF)

**S2 Fig. Area under the curve (AUC), average titer years (ATY) and width varying with age, using all tested strains.** Blue and red represent the AUC for the baseline and follow-up visit, respectively. Purple indicates the differences of indicators between the two visits. Solid lines are predictions from gam and the colored dashed lines represent the corresponding 95% confidence intervals. The sloping black dotted lines in panel J to L indicate the year of birth of participants. The dashed lines in panel J to L indicate the unweighted average isolation year of all strains.
(TIF)

**S3 Fig. Width of antibody profiles varying with age.** Widths were calculated using post-birth strains only. Panel A to C demonstrate width above titer 1:10, and Panel D to F demonstrate width above titer 1:40. Blue and red represent the indicators measured for serum collected in 2010 and 2014, respectively. Purple indicates the differences of indicators between the two visits. Solid lines are predictions from generalized additive model and the colored dashed lines represent the corresponding 95% confidence intervals. Results were calculated including all strains.
(TIF)

**S4 Fig. Odds of seroconversion by H3N2 strains.** Logistic regression models were fitted using age at sampling, prior titer and strains to predict the seroconversion. Coefficients for H3N2 strains are shown in the figure. The A/HongKong/1968 strain was set as reference.
(TIF)

**S5 Fig. Changes in titers to four recent strains.** (A) Distribution of changes in titers against recent H3N2 strains by the number of strains with increased titers. (B) Distribution of changes in titers against recent H3N2 strains by individual strain. We divided the changes in titers into four categories, i.e. decrease (green), no change (grey), two-fold increase (light purple) and four-fold change (seroconversion, dark purple).
(TIF)

**S6 Fig. Comparison of prediction performance of models including pre-existing immunity, assuming a linear effect of age.** Yellow and blue represents AIC and BIC, respectively. Dashed lines represent the AIC/BIC for models that only included titer to the examined strain $i$. Dotted lines represent the AIC/BIC for models that included titers to the examined strain $i$ and the prior strain $i-1$. Dots are AIC/BIC for models including additional predictor of pre-existing immunity of strains up to strain $i-1$.
(TIF)

**S7 Fig. Directed acyclic graphs of hypothesized relations between immune responses to past strains, titer to recent strain and seroconversion to recent strain.** Indirect effect (*path a path b*): immune responses to previous strains have positive association between titer to strain $i$ due to cross-reactions (*path a*), which has a negative association with seroconversion to strain $i$ (*path b*). Direct effect (*path c*): effect of immune responses on seroconversion to strain $i$ that was not mediated by titer to strain $i$. Total effect (*path a path b + path c*): combination of indirect effect and direct effect. Confounding effect (*path d*, *e* and *f*).
(TIF)

**S8 Fig. Mediation analysis of the effects of immune responses to previous strain on seroconversion to a recent strain.** Solid lines and filled squares represent the estimates from mediation analysis that did not consider interactions. Dashed lines and open circles represent the estimates from mediation analysis that considered interactions.
(TIF)

**S9 Fig. Changes in HAI titers between two visits.** Results are shown by subgroups of participants who had decreased (A), unchanged (B), increased (C) and four-fold increased (D) titers between the two visits, respectively.
(TIF)

**S10 Fig. Non-linear associations between age at sampling and seroconversion of four recent strains.** Models has been adjusted for titer to strain $i$, titer to strain $i$-$1$, and summary metrics.
(TIF)

**S11 Fig. Area under the curves (AUC), average titer years (ATY) and width varying with age, excluding participants who self-reported had been vaccinated against influenza.** Blue and red represent the AUC for the baseline and follow-up visit, respectively. Purple indicates the differences of indicators between the two visits. Solid lines are predictions from gam and the colored dashed lines represent the corresponding 95% confidence intervals. The sloping black dotted lines in panel J to L indicate the year of birth of participants. The dashed lines in panel J to L indicate the unweighted average isolation year of post-birth strains.
(TIF)

**S12 Fig. Association between pre-existing titer to individual strain and seroconversion to recent four strains.** Univariable coefficient (black) is estimated from univariable logistic regression of seroconversion to strain i on pre-existing titer to the strain listed in x-axis. Multivariable coefficient (red) is estimated from multivariable logistic regression of seroconversion to a strain $i$ on pre-existing titer to the strain listed in x-axis, adjusting for age at sampling and titer to strain $i$ and $i$-$1$.
(TIF)

**S13 Fig. Comparison of prediction performance of models including pre-existing immunity, assuming a non-linear effect of age.** Yellow and blue represents AIC and BIC, respectively. Dashed lines represent the AIC/BIC for models that only included titer to the examined strain $i$. Dotted lines represent the AIC/BIC for models that included titers to the examined strain $i$ and the prior strain $i$-$1$. Dots are AIC/BIC for models including additional predictor of pre-existing immunity of strains up to strain $i$-$1$.
(TIF)

**S14 Fig. Distribution of H3N2 strains by changes in titers between two visits.** We divided the examined data (i.e. all data, or pre-existing titer is greater or less than 1:80) on titer changes into four subgroups, i.e. decreased (green), unchanged (grey), any fold increase (light purple, including four-fold or more increase) and four-fold or more increase (dark purple). Colored points and lines represent the distribution of H3N2 strains within each subgroup. Colored bars represent the distribution of H3N2 strains regardless of titer changes for the examined data. (A) all data; (B) a subset contains pre-existing titers $\leq$ 1:40; (C) a subset contains pre-existing titers > 1:40. Insets D to F illustrate the distribution of changes in titers between two visits.
(TIF)

**S15 Fig. Association between pre-existing titer and seroconversion.** Univariable analysis of pre-existing titer on seroconversion. Coefficient was derived from univariable logistic regression of seroconversion to strain in x-axis on pre-existing titer to a strain listed in y-axis. Each cell represents an individual model. (B) Multivariable analysis of pre-existing titers on seroconversion. Coefficients were derived from multivariable logistic regression of seroconversion

to a strain in x-axis on age at sampling and pre-existing titers to all strains listed in y-axis. Each column represents an individual model. Each cell within a column represents the association between pre-existing titer to the strain listed in the y-axis on the seroconversion to strain in x-axis, after adjusting for the pre-existing titers to the rest of the twenty strains. Asterisks indicate $p \leq 0.01$.
(TIF)

**S16 Fig. Predicted probability of seroconversion and observed proportion of seroconversion by age group.** Models are fitted with a linear term on age, i.e. models used in Table 1. Age group was binned by 10 years. Horizontal lines represent the interquartile of predicted probability of seroconversion for the age group. Vertical lines represent 95% CI of the observed proportion of seroconversion derived from binomial distribution.
(TIF)

**S17 Fig. Predicted probability of seroconversion and observed proportion of seroconversion by age group, accounting for non-linear effect of age.** Models are fitted with a spline term on age, i.e. models used in S9 Table. Age group was binned by 10 years. Horizontal lines represent the interquartile of predicted probability of seroconversion for the age group. Vertical lines represent 95% CI of the observed proportion of seroconversion derived from binomial distribution.
(TIF)

**S1 Table. Comparison of demographic characteristics of participants.**
(DOCX)

**S2 Table. Geometric mean titer of tested H3N2 strains.**
(DOCX)

**S3 Table. Associations between pre-existing immunity and seroconversion to four recent strains, using titers to all tested strains.**
(DOCX)

**S4 Table. Associations between pre-existing average titer year or width above detectable threshold and seroconversion to four recent strains.**
(DOCX)

**S5 Table. Associations between and pre-existing immunity and seroconversion to four recent strains, considering the non-linear impact of age.**
(DOCX)

**S6 Table. Univariable logistic regressions of seroconversion to four recent strains on age and pre-existing immunity.**
(DOCX)

**S7 Table. Univariable analysis of predictors used to assess the association between pre-existing immunity and seroconversion to four recent strains.**
(DOCX)

**S8 Table. Comparison of demographic characteristics of participants who self-reported to have not been vaccinated against influenza.**
(DOCX)

**S9 Table. Associations between and pre-existing immunity and seroconversion to four recent strains, participants who reported never had been vaccinated against influenza.**
(DOCX)

**S10 Table. Associations between pre-existing immunity and seroconversion to four recent strains after accounting for sample collection time.**
(DOCX)

## Author Contributions

**Conceptualization:** Justin Lessler, Huachen Zhu, Chao Qiang Jiang, Jonathan M. Read, Kin On Kwok, Yi Guan, Steven Riley, Derek A. T. Cummings.

**Data curation:** Bingyi Yang, Justin Lessler, Ruiyin Shen.

**Formal analysis:** Bingyi Yang, Justin Lessler, Derek A. T. Cummings.

**Funding acquisition:** Justin Lessler, Chao Qiang Jiang, Jonathan M. Read, Yi Guan, Steven Riley.

**Investigation:** Huachen Zhu, Ruiyin Shen, Yi Guan.

**Methodology:** Bingyi Yang, Justin Lessler, Derek A. T. Cummings.

**Supervision:** Derek A. T. Cummings.

**Visualization:** Bingyi Yang, Justin Lessler, Derek A. T. Cummings.

**Writing – original draft:** Bingyi Yang, Justin Lessler, Derek A. T. Cummings.

**Writing – review & editing:** Bingyi Yang, Justin Lessler, Huachen Zhu, Chao Qiang Jiang, Jonathan M. Read, James A. Hay, Kin On Kwok, Ruiyin Shen, Yi Guan, Steven Riley, Derek A. T. Cummings.

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
