## [Decision Letter · Decision Letter 0]

10 Apr 2020

Dear Dr Yang,

Thank you very much for submitting your manuscript "Life course exposures continually shape antibody profiles and risk of seroconversion to influenza" for consideration at PLOS Pathogens. As with all papers reviewed by the journal, your manuscript was reviewed by members of the editorial board and by several independent reviewers. In light of the reviews (below this email), we would like to invite the resubmission of a significantly-revised version that takes into account the reviewers' comments.

I read the manuscript along with three reviewers. I share their concerns/comments and encourage the authors to take them onboard. In particular, Reviewer 1 highlights the very dense nature of the text. I think this paper was originally written for Science or Nature and this is still evident. It would be worth carefully expanding key topics. Reviewer 2 highlights important choices about the viruses that were included in the study and these issues should be addressed. Reviewer three raises interesting points about the timing of virus circulation during the lifetimes of study participants and these should be discussed in the main text.

We cannot make any decision about publication until we have seen the revised manuscript and your response to the reviewers' comments. Your revised manuscript is also likely to be sent to reviewers for further evaluation.

Sincerely,

Colin A. Russell

Guest Editor

PLOS Pathogens

Ron Fouchier

Section Editor

PLOS Pathogens

Kasturi Haldar

Editor-in-Chief

PLOS Pathogens

orcid.org/0000-0001-5065-158X

Michael Malim

Editor-in-Chief

PLOS Pathogens

orcid.org/0000-0002-7699-2064

Reviewer's Responses to Questions

**Part I - Summary**

Reviewer #1: Yang et al. analyze a large dataset of 777 paired serological samples to generate antibody profiles against 21 H3N2 strains, at two timepoints. The authors develop new metrics, nAUC, nW_z and nATY to measure the overall magnitude of antibody response, breadth of antibody response, and temporal center of mass of the antibody response, based on each individual’s antibody profile. Empirically, many of the study’s findings confirm results from past studies of antibody landscapes, or seroconversion, but the development of new metrics to quantify, analyze and simply talk about these complicated, multidimensional features of serological data is a significant step forward, and the importance of this work should not be taken for granted.

Overall, this is a thoughtful and well-written study. The authors have clearly done a lot of careful analysis and manage to present their findings with impressive clarity and conciseness, given how complicated it is to work with and interpret longitudinal serological data. I have a few comments, but overall, I found the study quite impressive.

Reviewer #2: The manuscript describes innovative methods to quantify how antibody titres and titre rises across multiple temporally spaced influenza A(H3N2) virus strains differ between groups, in this case looking at age groups. The study was well designed to collect samples a from relatively large numbers of participants aged 2-86 Y at two time-points ~ 4-years apart, and measured serum HI antibody titres against 21 A(H3N2) viruses representing ~ 46 years of virus evolution in humans (1968 – 2014). Antibody profiles were constructed by plotting each individuals HI titres against the 21 strains ordered chronologically. The main aim of the study is to derive metrics that may be used to summarize antibody profiles at different times, and then determine how profile metrics at time 1 may impact seroconversion and titre rise. Three equations or statistics were devised to characterize antibody profiles: “height” (AUC); “width” (summed time-intervals over which titres exceeded particular thresholds); and “temporal targeting of antibodies” (average year where antibody titres are concentrated). It was clear that people aged 40-60 years had relatively restricted antibody profiles in terms of AUC and width, which were centered on strains present around the midpoint of life, but it was less clear how these metrics impacted subsequent seroconversion and titre rise. Many models and tables were presented to show that some metrics improved models, albeit modestly compared to the effect of baseline homologous titre. However, the abstract, results and discussion do not clearly convey how important metrics were in terms of effects on immune responses.

Overall this is a nice study but the manuscript needs substantial re-writing, using more concise language, to convey the results and their meaning more effectively. Particular attention should be paid to the abstract. In addition, the supporting information is not set out in any particular order. The description of the supporting information does not follow the numbering of the figures and tables. It would be easier to follow the supporting information if it was broken into sections by the issue being addressed (E.g. Effects of normalization; Factors affecting seroconversion…., with the relevant supporting text, figures and tables adjacent to each other), and if there was some text to help the reader interpret the results that are being presented in each piece of supporting information. The manuscript may also be improved by carefully re-considering which topics to focus on and include in the main manuscript rather than in the supporting information.

Reviewer #3: In this paper, authors collected serum samples of people aged 2 to 86 at two time points. The baseline sampling was done in 2009 to 2011 and follow-up was done in 2014-2015. As a result, 777 paired hemagglutination inhibition titers against 21 H3N2 strains were obtained. Antibody profiles of participants were generated and used to construct models that predict the risk of infections to recent strains. This study proposed three new metrics, AUC, Wz, ATY, aiming to determine antibody profile variation for predicting the risk of seroconversion. Using logistic regression model, effects of these metrics on seroconversion were discussed. However, the usefulness of the proposed models in predicting seroconversion is unclear.

**Part II – Major Issues: Key Experiments Required for Acceptance**

Reviewer #1: 1. My primary big-picture comment is that as currently written, the manuscript does not take nearly enough advantage of the flexible formatting guidelines provided by PLOS Pathogens. The text is certainly well written, but incredibly dense, possibly to the point of being difficult to follow for a non-influenza expert. Some of the main text figure panels (e.g. 2D-I) are rich with information, but not presented in the results section at all, which is a shame, and some of the results that are presented could be clearer with a bit more elaboration. It also seems a shame that the mediation analysis (Fig. S12) isn’t mentioned until the Discussion, and buried deep in the Supplement.

2. Can the authors elaborate a bit more on the fact that titers increased across the board, even to non-recent strains, from baseline to follow up (as shown clearly in Fig. 2)? This is presented as a result, and attributed to recent exposures. But the pattern is striking, and alternate explanations involving longer storage of the baseline samples, or storage issues affecting only the baseline samples, haven’t been discussed. The results also leave me wondering if the baseline and follow-up samples were collected at different times of year, before and after the start of influenzas season, respectively, so that they can be thought of as analogous to acute and convalescent samples?

Edit: I’m only just now (after having read the study more than once), realizing that the baseline samples were collected shortly after the 2009 pandemic, at a time when H3N2 was not the dominant circulating subtype, and H3N2 titers had not been boosted in a while. Presumably there was a big H3N2 season prior to collection of most of the follow-up samples? This is an important point, which makes the observation that titers rose across the board make a lot more sense, and which should be made explicit in the text.

3. The biological interpretation of nATY is lines 153-157 is very lucid and well-written. But I’m worried that this passage conflates the extreme case, presented for illustration (nATY tracks birth year), with a biological expectation. This extreme case could only come true if individuals developed and boosted titers only to the strain of first exposure, and were never again in their lifetimes able to generate a de novo response to new strains. We know empirically this is not how influenza immunity works, and I don’t think anyone in the field would really expect nATY to track birth year perfectly—the fact that nATY moves away from birth year a little bit each year (as people are exposed to new strains and develop some de novo responses), isn’t totally surprising and I’m not sure that this in itself logically rules out any alternatives. It would be great if the authors could clarify the biological expectations here, and elaborate a bit more on the observed patterns—does nATY asymptotically approach the unweighted average over time, or does it always remain somewhere between birth year and the unweighted average. Does nATY ever skew higher than the unweighted average?

A related point, which would lay the questions above to rest: Is the black dashed line showing unweighted average isolation year missing in Fig 3 J-L? Same comment in Fig. S10. I also see this dashed line is present in Fig. S2 J-L, but I’m not sure it’s plotted correctly…here I see a flat line, but if the unweighted average only considers post-birth strains, I don’t think it should be the same for all ages.

Reviewer #2: 1. Timing of sample collection. Given that this study investigates factors that affect antibody titre change, the timing of sample collection, and how this may differ between individuals, could be very important, and should be taken into account. The periods over which samples were collected at baseline (2009-2011) and follow-up (2014-2015) are both long, such that potential virus exposure may differ between individuals and introduce experimental differences between antibody titre and titre-rise profiles.

2. Viruses

2a. The methods should describe how were viruses were propagated – eggs or cells and cell-type?

2b. Clade 3c.2a viruses such as HongKong/14 agglutinate turkey red blood cells very poorly and hence assays sensitivity may be poor. Was there any validation to determine whether this could account for relatively low titres against this virus in this study? In addition, some recent viruses can agglutinate via neuraminidase, which in turn can impair detection of HI antibodies. Were virus HA and NA genes sequenced and were titres tested with and without oseltamivir to determine whether titres could be affected by NA-agglutination.

2c. Clade 3c.3a viruses (Eg. A/Switzerland/9715293/13) circulated widely and were associated with large epidemics in 2014, so should be included.

3. Models to understand antibody profile effects on seroconverion. Why was age effect considered to be linear in the main analysis when Figure 3 indicates that effects will not be linear? Why is i-1 strain titre included in all models when this had effects on seroconversion against A/Perth/2009 but not other strains. Could the situation with regard to i-1 strain reveal more complex interactions that depend on the strains involved? How do the authors interpret the relative impacts of AUC, width and ATY, other than that they were all positively associated with seroconversion? Are they all independently important or just related factors that indicate whether or not there is substantial existing immunity which provided a basis for boosting? There is some attempt to understand boosting versus interference (and ceiling) effects in relation to the opposite(?) effects of homologous-strain versus prior strain titres but this is not explained in much detail (line 226-229) and is hard to follow.

Reviewer #3: Possible effect of H1N1 epidemics should be discussed. In the panels G and H in Figure 2, there is a "saddle" where we can see low titer to the H3N2 strains isolated during 1975-1980. This low titer might be related to H1N1 pandemic of the Russian flu in 1977. This point can be discussed in somewhere in the Discussion along with the concept of original antigenic sin or antigenic seniority.

For those who were born before 1968, there is no or less chance to be exposed to H3N2 viruses in their early life compared to those born after 1968. In Figure 3J and 3K, nATY is continuously declining up to the age around 50. The decrease in nATY stopped the age around 50 maybe because of no exposure to H3N2 viruses. However, in page Page 9, Line 157-158, authors wrote “Our empirical observations are more in line with the latter hypothesis”, which assumes that "all strains circulating in one’s life were equally important". This description should be modified and the possible interpretation should be added.

The authors claim that "we found that participants who had higher immunity to previously exposed strains were more likely to experience seroconversion to recent strains after adjusting for homologous titer" in Page 12, Line 221-223. However, it is not clear how Figure S11 and Table S10, S11 support this statement. The authors should clarify this point.

**Part III – Minor Issues: Editorial and Data Presentation Modifications**

Reviewer #1: 1. Overall, the writing in the manuscript is excellent, clear and easy to read. The Authors Summary, however, is a bit choppy in comparison to the rest of the text—consider text edits to tighten this up on revision.

2. At their core, AUC and W_z seem to present different ways to measure the overall magnitude of an individual’s antibody response. (i.e. how much of a high-responder is any given subject?) Is there any case where we should expect these quantities to diverge in the same individual? After developing and working with them, do the authors feel one is more useful than another?

Reviewer #2: 1. The inclusion of A/HongKong/2014 as a recent strain is questionable since there may have been limited circulation of that virus prior to follow-up sampling in 2014-2015.

2. The interpretation of width is somewhat limited due to the imposition of thresholds for defining the strains against which a response is detected. Can this metric be additionally weighted by titre for each strain? For example if the titre is 20 instead of 40 the width is increased by half that time interval instead of the full time interval.

3. Multiple statements in the abstract are ambiguous as follows:

Line 38 – should qualify the time period (~ 4 Y) over which 97.8% had a titre rise.

Line 40 – what does “recent strains exhibited the greatest variation” mean? Greatest titer rise,variation between age groups?

Line 41 – qualify “adjusting for homologous titer” = at baseline?

Line 42 - qualify = seroconversion against recent strains?

4. Results - Some of the key findings presented relate to deficits in antibody profiles among people aged 40 – 60 years. In this regard, it would improve the manuscript if additional profiles for individuals from this age group are presented in Figure 1 or as supplementary information.

Figure 1 – it may help to tie the manuscript together if AUC, width, and ATY values are also indicated for each individual shown.

Figure 2 – as above, it may help to tie results together if age, AUC, width and ATY are indicated for the individuals shown as examples (panels A, E, I).

Line 111 – referring to protective titres is controversial. Although 1:40 titres may be protective, or be associated with a 50% reduction in the risk of infection, this is not universal. It would be better to just say that 95.6% of participants had titres of at least 1:40 to at least one strain, and let the readers decide what that may mean.

Line 112: 116 – it would be more meaningful to present GMT for post-birth, rather than all strains to understand differences between time 1 and time 2, and again interpretation of this result will depend upon the time of sample collection relative to virus transmission in the community.

Line 117 – how was the age at time of isolation of the strain against which titers were highest determined from the multiple GAM fits? Can the GAM model R code be included in the supplement?

Line 147-152 – This is very hard to follow. What is ratio to peak? Width is described in terms of percentage of strains against which titres were above threshold, whereas on line 149 the authors switch to fraction.

Figure S6 and S7 are included to justify that models to predict seroconversion are improved if titre to strain i, strain i-1 (variable improvement), AUC and W40 are all included, but what is the difference between Figure S6 and S7. It is clear that the values differ but there is no explanation as to why?

– Consider substituting the term “risk” for capacity- or odds- of seroconversion to avoid the inference that factors such as a high AUC are a risk factor for being infected, when what you are really looking at is the capacity to produce antibodies.

5. Discussion

Line 210 – the authors suggest that antibody nautch and nWidth are relatively low among people in their 40-50s because other immune responses, which are not being measured, are preventing infections from occurring. However, their data indicates that 97.8% of people are having a titre rise which they surmise to represent infection. Please justify this inference considering the data presented, as well as data from other studies on A(H3N2) virus infection rates in different age groups.

Line 233 – positive effects of titres to old strains on seroconversion against recent strains were only detectable in multivariate analysis.

6. Supporting Information

– There are many tables (S3, S4, S5, S6, S7) with different versions of models, mentioned in different parts of the manuscript. This makes it quite hard to follow what is going on, and what differs between models.

Fig S4 – The caption should state that Odds Ratios were for comparison with seroconversion against HK/1968. It is not sufficient to state this in the main text only.

Fig S5 – what do the colours mean?

7. Minor typographical or grammatical errors

Line 372 – the each

Line 377 – decrease should be decreased

Line 603 –among for

Line 616 -618 - of people aged 40-50 years (xxxx) of participants who

Line 644 – When fitted logistic regressions of seroconversions on recent strains with age at ….(rewrite this sentence).

Line 656 – antigenically relatives

Reviewer #3: Page 5. Line 70-71:

Add reference to other works such as

Nachbagauer R, Choi A, Hirsh A, et al. Defining the antibody cross-reactome directed against the influenza virus surface glycoproteins. Nat Immunol. 2017;18(4):464–473. doi:10.1038/ni.3684

Page 6. Line 95:

The term seroconversion should be defined precisely. Use an independent sentence to define what is seroconversion.

Page 7. Line 107-108:

Year 2009 has two strains (i.e. A/Perth/2009, A/Victoria/2009). The reason why two strains were included should be explained.

Page 7. Line 110:

(Fig 1, A and C) should be (Fig 1, A-C), because Fig 1 B also contains "pre-birth strains".

Page 7. Line 113:

GMTs of pre-birth strains should be shown in Table S2.

Page 7. Line 116-117:

The authors wrote, “Participants had the highest titer to strains that were isolated within the first decade of their life (4.3 years; IQR, 2.0 to 6.9 years across strains) (Fig. S1)”. However, it is difficult confirm this description only form Fig. S1. Is it possible to have another representation to visualize this claim? It is also not clear from which table the median and IQR were derived.

Page 7. Line 117-118:

It is not clear from which Table the statistics of GMTs of recent strains and that of non-recent strains for baseline and follow-up visits were derived. It seems to be Table S2. If so, the table should be referred in the body text.

Page 8. Line 136:

"Non-normalized analysis included in SI Materials and Methods" should be "...in S1 Appendix".

Page 9. Line 143:2.1 (95% CI, 1.9 to 2.4) and 1.7 (95% CI, 1.5 to 1.9)

The method used for the calculation of these confidence intervals of odds ratio should be clarified.

Page 9. Line 162:

In the statement "Fig. 2C and f", "F" should be in capital letter.

Page 9. Line 163:

The author wrote "73.7% showed a 4-fold or greater titer increase (seroconversion) to one or more (Fig. 2I)". However, it is difficult to confirm this proportion from Fig. 2I.

Page 11. Line 190-193:

The sentence "Based on the impact of seroconversion and transient antibody dynamics..." is difficult for readers to understand. It would be better to break it down into smaller and simpler sentences to describe the findings.

Page 11. Line 194:

Fig. 2F could not demonstrate that the increased antibody titers were those against non-recent strains. It may be better to refer to Fig. 2C instead.

Page 12. Line 211:

Grammatical error of the phrase ".., could preventing people...".

Page 12. Line 229:

"detailed in S Appendix" should be "detailed in S1 Appendix"

Page 14. Line 712:

The author mentioned "Table S18". This should be "Table S8"

Page 16. Line 295:

There should be a space between "A/Wuhan/1995," and "A/Victoria/1998".

Page 17. Line 327:

The values are added from i=1 to M-1. It is not clear why you stop at M-1 instead of M. Please describe the reason. Is it related to the two strains in 2009?

Page 20. Line 369: GAM

Detailed information of the model is needed, i.e. the distribution and link function used in GAM.

Page 24. Line 441-442:

Citation of Francis T 1960 contains its title twice.

Page 33. Table 1:

The definitions of Model 1, 2, and 3 are not clear. Please describe which are Model 1, 2, and 3 in the Method Section. Explanations on how to interpret values are needed for general readers. For instance, what does it mean when values are larger than 1 or smaller than 1.

Page 38. Line 659:

"per-existing" should be "pre-existing".

Page 41. Line 716:

Grammatical error of the phrase "Vaccination status of influenza seems not affect...".

Page 45. Line 747-749:

It seems that the coefficient was normalized by comparing to A/HongKong/1968 as written in Line 165 in Page 10. If so, the figure legend should be collected to indicate this.

Page 46. Line 753:

"changes in tiers" should be "changes in titers".

Page 46. Fig. S5.

There is no explanation for the different colors in this figure.

Page 47-48. Fig. S6-S7:

Explain the difference between Figure S6 and S7. The Figures look different but AIC and BIC values are the same. Figure legends are also the same.

Page 52. Fig. S11b:

Figure legend is not clear. If each column represents an individual model as stated, it is unclear what each row within the column means.

Page 61. Table S1:

It is not clear what Pa means.

Supplementary Information:

There are so many supplementary figures and tables. The reviewer recommends providing only figures and tables essential for this manuscript.

PLOS authors have the option to publish the peer review history of their article (what does this mean?). If published, this will include your full peer review and any attached files.

Reviewer #1: No

Reviewer #2: No

Reviewer #3: No
---

## [Editor Report · Decision Letter 1]

14 May 2020

Dear Dr Yang,

We are pleased to inform you that your manuscript 'Life course exposures continually shape antibody profiles and risk of seroconversion to influenza' has been provisionally accepted for publication in PLOS Pathogens.

Best regards,

Colin A. Russell

Guest Editor

PLOS Pathogens

Ron Fouchier

Section Editor

PLOS Pathogens

Kasturi Haldar

Editor-in-Chief

PLOS Pathogens

orcid.org/0000-0001-5065-158X

Michael Malim

Editor-in-Chief

PLOS Pathogens

orcid.org/0000-0002-7699-2064
---

## [Editor Report · Acceptance letter]

19 Jun 2020

Dear Dr Yang,

We are delighted to inform you that your manuscript, "Life course exposures continually shape antibody profiles and risk of seroconversion to influenza," has been formally accepted for publication in PLOS Pathogens.

Best regards,

Kasturi Haldar

Editor-in-Chief

PLOS Pathogens

orcid.org/0000-0001-5065-158X

Michael Malim

Editor-in-Chief

PLOS Pathogens

orcid.org/0000-0002-7699-2064